# Divergent projections of the prelimbic cortex bidirectionally regulate active avoidance

**Maria M Diehl[†]\*, Jorge M Iravedra-Garcia, Jonathan Morán-Sierra, Gabriel Rojas-Bowe, Fabiola N Gonzalez-Diaz, Viviana P Valentín-Valentín, Gregory J Quirk**

Departments of Psychiatry and Neurobiology & Anatomy, University of Puerto Rico School of Medicine, San Juan, Puerto Rico

**Abstract** The prefrontal cortex (PFC) integrates incoming information to guide our actions. When motivation for food-seeking competes with avoidance of danger, the PFC likely plays a role in selecting the optimal choice. In platform-mediated active avoidance, rats avoid a tone-signaled footshock by stepping onto a nearby platform, delaying access to sucrose pellets. This avoidance requires prelimbic (PL) PFC, basolateral amygdala (BLA), and ventral striatum (VS). We previously showed that inhibitory tone responses of PL neurons correlate with avoidability of shock (Diehl et al., 2018). Here, we optogenetically modulated PL terminals in VS and BLA to identify PL outputs regulating avoidance. Photoactivating PL-VS projections reduced avoidance, whereas photoactivating PL-BLA projections increased avoidance. Moreover, photosilencing PL-BLA or BLA-VS projections reduced avoidance, suggesting that VS receives opposing inputs from PL and BLA. Bidirectional modulation of avoidance by PL projections to VS and BLA enables the animal to make appropriate decisions when faced with competing drives.

\*For correspondence:
maria.m.diehl@gmail.com

Present address: [†]Department of Psychological Sciences, Kansas State University, Manhattan, United States

Competing interests: The authors declare that no competing interests exist.

## Introduction

The prefrontal cortex (PFC) controls goal-directed behaviors by integrating sensory, motor, and memory information to guide an individual's actions (*Fuster, 1997*; *Miller and Cohen, 2001*; *Kesner and Churchwell, 2011*). A leading hypothesis is that the PFC evaluates possible outcomes in the presence of competing drives (*Ridderinkhof et al., 2004*; *Rushworth and Behrens, 2008*; *Bissonette et al., 2013*; *Saunders et al., 2017*), such as seeking food rewards in the presence of a potential threat (*Aupperle et al., 2015*; *Friedman et al., 2015*; *Schumacher et al., 2016*; *Bublatzky et al., 2017*). This situation has been modeled in rodents using the platform-mediated active avoidance (PMA) task, in which food-restricted rats learn to avoid a tone-signaled shock by moving to a nearby platform, at the cost of delaying access to sucrose pellets (*Bravo-Rivera et al., 2014*). Activity in the prelimbic (PL) PFC is necessary for the expression of PMA (*Bravo-Rivera et al., 2014*; *Diehl et al., 2018*), but little is known about the circuit-level mechanisms of the PFC that control this type of avoidance.

During PMA, neurons in PL show both excitatory and inhibitory responses to the tone (*Diehl et al., 2018*). Similar excitatory tone responses in PL are observed in fear-conditioned rats (*Burgos-Robles et al., 2009*), but inhibitory tone responses are limited to rats trained in PMA, suggesting the existence of a disinhibitory PL circuit for expression of avoidance (*Diehl et al., 2018*). PL projects heavily to the ventral striatum (VS) and the basolateral amygdala (BLA) (*Sesack et al., 1989*; *Vertes, 2004*), both of which are necessary for PMA (*Bravo-Rivera et al., 2014*) as well as other types of active avoidance (*Darvas et al., 2011*; *Ramirez et al., 2015*).

We therefore sought to determine the role of PL projections to VS and to BLA in PMA using optogenetic techniques to identify possible targets of avoidance-related PL activity. Following training in PMA, PL projections to VS or BLA were either photoactivated (with channelrhodopsin) or photosilenced (with archaerhodopsin). We also used this method to assess the role of BLA projections to VS. We suggest that PL controls active avoidance by bidirectionally modulating outputs to BLA and VS.

## Results

### PL projections to the VS: photoactivation impairs avoidance

Our previous study revealed that inhibitory tone responses in PL neurons were only observed in rats that received avoidance training (*Diehl et al., 2018*). We interpreted this as inhibitory responses in PL signaling the avoidability of the tone-predicted shock. Opposing these inhibitory responses by photoactivating PL at the baseline rate of 4 Hz impaired avoidance. Because PL projects densely to the VS (*Sesack et al., 1989*; *Vertes, 2004*), we hypothesized that inhibitory responses in PL neurons projecting to the VS would promote avoidance. If so, activating these projection neurons with channelrhodopsin (ChR2) would be expected to impair avoidance. Following viral infusion and surgical implantation of optic probes, rats were trained in PMA over 10 days as previously described (*Figure 1A*, *Bravo-Rivera et al., 2014*; *Rodriguez-Romaguera et al., 2016*; *Diehl et al., 2018*). Histological analysis showed that expression of ChR2 was largely confined to rostral PL with some spread to rostral anterior cingulate (Cg1) and caudal PL (*Figure 1—figure supplement 1A*). Following training, blue laser light (473 nm) was used to activate PL terminals in VS, at a frequency of either 4 Hz or 15 Hz (*Figure 1B*).

*Figure 1C* shows that ChR2 photoactivation at either frequency significantly reduced the time spent on the platform during the tone, compared to eYFP controls (4 Hz: eYFP 77% vs. ChR2 51%, $t_{(18)}$=2.370, p>0.05, Bonferroni corrected; 15 Hz: eYFP 88% vs. ChR2 22%, $t_{(18)}$=7.152, p<0.01, Bonferroni corrected). Analysis of avoidance across the tone in 3 s bins (*Figure 1D*) indicated that ChR2 rats were significantly delayed in their avoidance 3–6 s after tone onset during 4 Hz photoactivation (repeated-measures ANOVA, $F_{(1,9)}$ = 5.88, p=0.026; post-hoc Tukey test, 3–6 s p=0.029). Photoactivation at 15 Hz had a stronger effect, with ChR2 rats showing a significant reduction of avoidance throughout the tone (*Figure 1E*, repeated-measures ANOVA, $F_{(1,9)}$ = 48.92, p<0.001; post-hoc Tukey test, 6–30 s, all p's < 0.01). Photoactivation of PL-VS projections had no effect on locomotion, as indicated by distance traveled in an open field (4 Hz; eYFP n = 15, 2.7 m vs. ChR2 n = 7, 2.2 m, $t_{(20)}$=1.54 p=0.138), nor on anxiety levels, as both groups spent a similar amount of time in the center of the open field (4 Hz: eYFP n = 15, 4.1 s vs. ChR2 n = 7, 6.4 s, $t_{(20)}$=1.17, p=0.255). Thus, photoactivation of PL terminals in VS during the tone impaired the expression of avoidance, similar to photoactivation of PL somata (*Diehl et al., 2018*).

If photoactivation impairs avoidance, we hypothesized that photosilencing PL-VS projections would enhance avoidance, similar to what we observed when photosilencing PL somata (*Diehl et al., 2018*). Archaerhodopsin (ArchT) was infused into PL and optic fibers were implanted to target PL terminals in VS (*Figure 1F*; *Figure 1—figure supplement 1B*). However, photosilencing PL-VS projections had no significant effect on average avoidance levels (*Figure 1G*, eYFP 76% vs. ArchT 69%, $t_{(20)}$=0.8498, p=0.406) or on the timecourse of avoidance across the tone (*Figure 1H*, repeated-measures ANOVA, $F_{(1,9)}$=0.7088, p=0.410). Photosilencing PL-VS projections also had no effect on spontaneous bar-pressing (ArchT, n = 9, seven average number of presses during laser OFF vs. six average number of presses during laser ON, $t_{(8)}$=1.32, p=0.224). These negative findings of photosilencing PL-VS projections disagree with our prior somatic results (*Diehl et al., 2018*) and may be due to a ceiling effect or other factors (see Discussion).

### PL projections to the BLA: photoactivation facilitates avoidance

Our previous study suggested that PL excitation was not necessary for avoidance, because photosilencing PL somata did not impair avoidance (*Diehl et al., 2018*). Nevertheless, excitatory responses of PL neurons were observed during both tone onset and platform entry, and the opposing effects on behavior of inhibitory and excitatory responses may have been masked by our non-specific targeting of PL somata. Anatomical studies demonstrate strong projections from PL to BLA

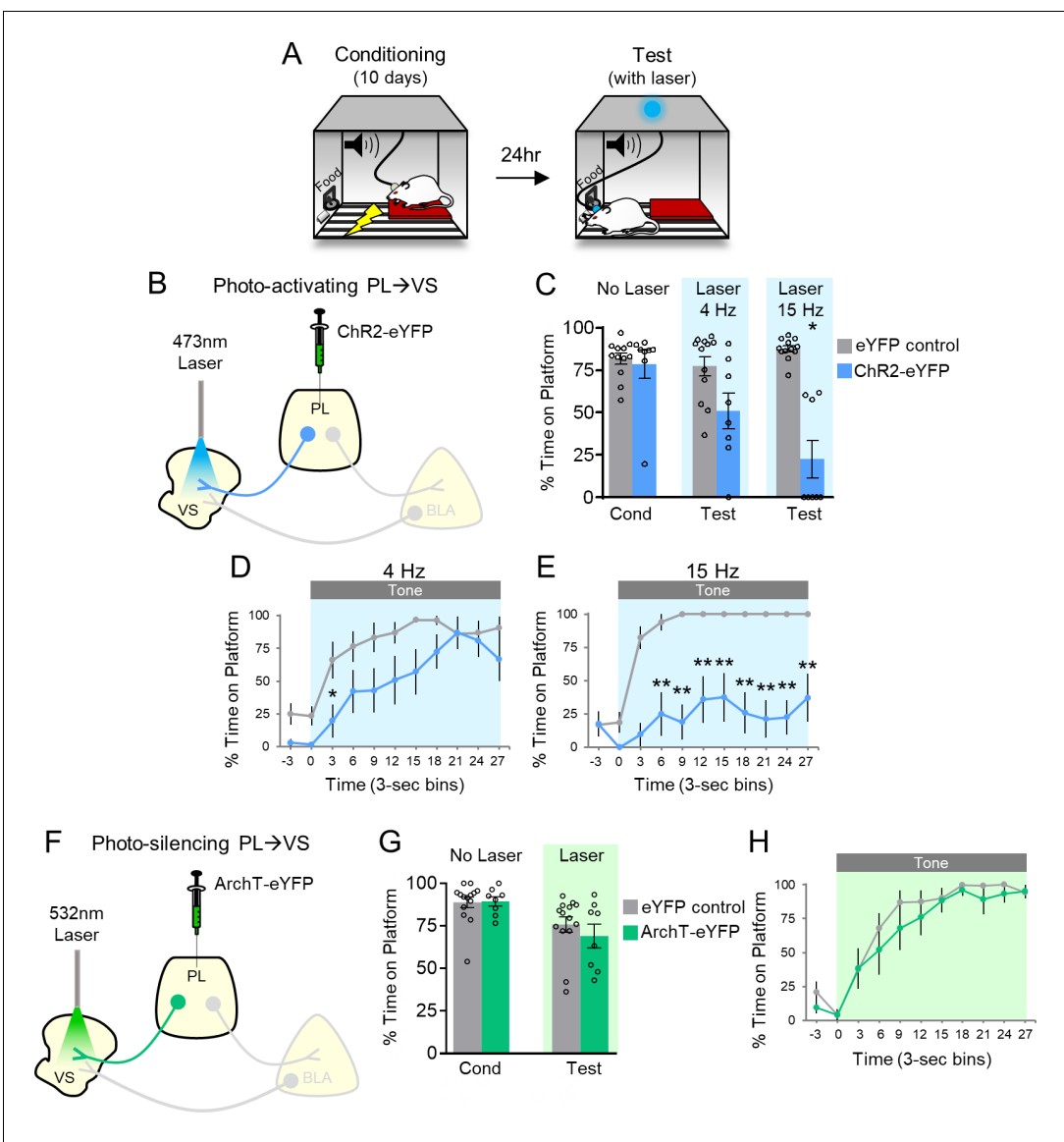

**Figure 1.** Photoactivation of PL projections to VS impairs avoidance. (**A**) Rats were trained in platform-mediated avoidance (PMA) for 10 days, followed by a test with laser illumination during the tone. (**B**) Schematic of virus infusion and optic probe placement. The CaMKII-α promoter was used for all AAVs. (**C**) Percent time on platform during the last day of avoidance conditioning (Cond; No Laser), and 4 Hz and 15 Hz Laser tests performed one or four days later. (**D**) Timecourse of avoidance during 4 Hz Laser revealed that PL-VS ChR2 (n = 8) rats showed delayed avoidance compared to eYFP (n = 12) controls (repeated-measures ANOVA, post-hoc Tukey). (**E**) Same as in panel D with 15 Hz Laser. (**F**) Schematic of ArchT virus infusion and optic probe placement. (**G**) Percent time on platform during the last day of avoidance conditioning (Cond; No Laser), and Test day (Tone with Laser). (**H**) Timecourse of avoidance during Test revealed that ArchT (n = 8) rats showed similar percent time on platform compared to eYFP (n = 14) controls (NS; repeated-measures ANOVA, post-hoc Tukey). All data are shown as mean ± SEM. *p<0.05, **p<0.01.

The online version of this article includes the following source data and figure supplement(s) for figure 1:

**Source data 1.** Percent time on platform during manipulation of PL-VS projections.
**Figure supplement 1.** Spread of AAV expression and location of optic probes.

(*Sesack et al., 1989*; *Vertes, 2004*). Therefore, we sought to determine if optogenetic manipulation of PL projections to BLA would modulate avoidance. ChR2 was infused into PL and optic fibers were implanted in BLA (*Figure 2A*; *Figure 2—figure supplement 1A*). In contrast to PL-VS photoactivation, PL-BLA photoactivation at 4 or 15 Hz did not impair avoidance; in fact, avoidance significantly increased with 15 Hz photoactivation (*Figures 2B*, 4 Hz: eYFP 80% vs. ChR2 90%, $t_{(11)}=1.652$, p=0.127; 15 Hz: eYFP 75% vs. ChR2 88%, $t_{(15)}=2.700$, p<0.05, Bonferroni corrected). Analysis across the tone (*Figure 2C–D*) revealed increased avoidance with 15 Hz photoactivation at both 3 s and 6 s time bins (Mann-Whitney U-test, 3–6 s p=0.006, 6–9 s p=0.027).

The small effect of photoactivation of PL-BLA projections is likely due to ceiling levels of avoidance, as control rats avoided at 82% (*Figure 2B*, No Laser Cond). To reduce avoidance levels, we extinguished a subset of rats by administering 15 trials of avoidance extinction (tone without shock, no laser, *Figure 2E*). On the last trial of extinction, photoactivation of PL terminals in BLA (at 15 Hz) significantly increased avoidance compared to eYFP controls (eYFP 2.4% vs. ChR2 34.5% time on

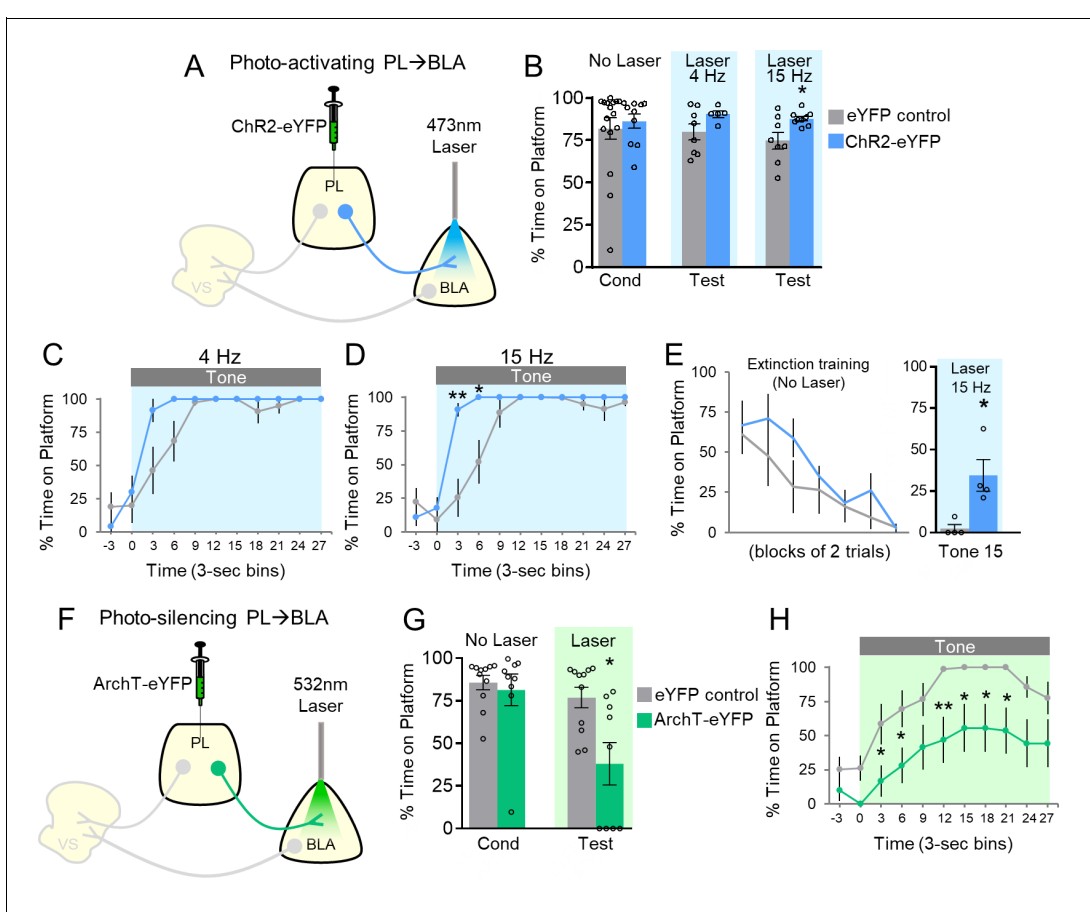

**Figure 2.** Avoidance requires activation of PL projections to BLA. (**A**) Schematic of ChR2 virus infusion and optic probe placement. (**B**) Percent time on platform during the last day of avoidance conditioning (Cond; No Laser), and 4 Hz and 15 Hz Tests (with Laser). (**C**) Timecourse of avoidance during the 4 Hz Test (ChR2 n = 5, eYFP n = 8). (**D**) Same as in panel B with 15 Hz Laser (ChR2 n = 9, eYFP n = 8). (**E**) Photoactivation of PL-BLA projections (15 Hz) reinstates avoidance following extinction (ChR2 n = 4, eYFP n = 4). (**F**) Schematic of ArchT virus infusion and optic probe placement. (**G**) Percent time on platform during the last day of avoidance conditioning (Cond; No Laser), and Test (Tone with Laser). (**H**) Timecourse of avoidance during Test revealed that ArchT (n = 9) rats showed delayed avoidance compared to eYFP (n = 11) controls (repeated-measures ANOVA, post-hoc Tukey). All data are shown as mean ± SEM. *p<0.05, **p<0.01.

The online version of this article includes the following source data and figure supplement(s) for figure 2:

**Source data 1.** Percent time on platform during manipulation of PL-BLA projections.
**Figure supplement 1.** Spread of AAV expression and location of optic probes.

platform, $t_{(6)}$=3.269, p=0.017). Photoactivation of this pathway did not affect locomotion (15 Hz: n = 6 eYFP, 3.2 m vs. n = 5 ChR2, 3.1 m of distance traveled in an open field, $t_{(9)}$=0.196, p=0.849), or anxiety levels (15 Hz: n = 6 eYFP, 5.1 s vs. n = 5 ChR2, 4.0 s of time spent in center of open field, $t_{(9)}$=0.318, p=0.757;). Overall, these findings suggest that PL activation of BLA promotes avoidance.

### PL projections to the BLA: photosilencing impairs avoidance

We next determined whether photosilencing PL-BLA projections would impair avoidance, demonstrating the necessity of this excitatory pathway in avoidance expression. Rats infused with ArchT in PL and implanted with optic fibers in BLA (*Figure 2F*; *Figure 2—figure supplement 1B*) underwent avoidance training followed by a test of avoidance expression. Photosilencing PL-BLA projections significantly impaired the expression of avoidance (*Figure 2G*, eYFP 77% vs. ArchT 38% time on platform, $t_{(18)}$=2.985, p<0.05, Bonferroni corrected). Furthermore, ArchT rats showed an overall decrease in avoidance across the tone compared to eYFP controls (*Figure 2H*, repeated-measures ANOVA, $F_{(1,9)}$=9.449, p=0.007, post-hoc Tukey tests, 3–9 and 12–24 s, all p's < 0.05). Photosilencing PL-BLA projections had no effect on spontaneous bar-pressing (ArchT n = 11, 4.82 average number of presses during laser OFF vs. 5.77 average number of presses during laser ON, p=0.255, $t_{(10)}$=1.21). These findings suggest that PL excitation of BLA is necessary for the expression of active avoidance.

### BLA projections to the VS: photoactivation facilitates avoidance

Thus far, distinct PL projections appear to have opposite roles in avoidance: PL-VS projections inhibit avoidance whereas PL-BLA projections promote avoidance. How might signaling between VS and BLA regulate avoidance? Previous studies have shown that BLA sends strong projections to VS (*Kelley et al., 1982*; *Mcdonald, 1991a*; *Mcdonald, 1991b*), and pharmacological disconnection of BLA and VS impairs shuttle avoidance (*Ramirez et al., 2015*). Therefore, we reasoned that BLA-VS projections may also promote avoidance. Following ChR2 infusion into BLA and implantation of optic fibers targeting the VS (*Figure 3A*; *Figure 3—figure supplement 1A*), photoactivation of BLA-VS projections slightly increased avoidance, but this did not reach statistical significance (*Figure 3B*) at 4 Hz (eYFP 70% vs. ChR2 90% time on platform, $t_{(12)}$=1.678, p=0.119) or 15 Hz (eYFP 70% vs ChR2 86% time on platform, $t_{(12)}$=1.50, p=0.159). There was also no significant difference in the timecourse of avoidance across the tone (*Figure 3C–D*).

Similar to photoactivation of PL-BLA projections, a ceiling effect was observed during photoactivation of BLA-VS projections. Therefore, a subset of rats underwent avoidance extinction to determine if photoactivating BLA-VS projections would reinstate avoidance expression (*Figure 3E*). In the final trial of extinction training, photoactivation of BLA-VS projections at 15 Hz significantly increased avoidance expression (eYFP 9% vs. ChR2 38% time on platform, $t_{(11)}$=3.521, p=0.005). Photoactivation did not affect locomotion (15 Hz: n = 5 eYFP, 2.7 m vs. n = 7 ChR2, 3.2 m of distance traveled in an open field, $t_{(10)}$=1.10, p=0.298), or anxiety levels (15 Hz: n = 5 eYFP, 4.0 s vs. n = 7 ChR2, 4.2 s of time spent in center of open field, $t_{(10)}$=0.0085, p=0.934). Taken together, these findings show that BLA-VS projections facilitate the expression of avoidance.

### BLA projections to the VS: photosilencing impairs avoidance

To establish the necessity of the BLA-VS excitatory pathway in avoidance expression, we assessed whether photosilencing BLA-VS projections would impair avoidance. Following infusions of ArchT into BLA and optic fiber implantation into VS (*Figure 3F*; *Figure 3—figure supplement 1B*), rats underwent PMA training followed by a test of avoidance expression. Photosilencing BLA-VS projections significantly impaired the expression of avoidance (*Figure 3G*, eYFP 78% vs. ArchT 28%, $t_{(19)}$=3.849, p<0.01, Bonferroni corrected). Furthermore, ArchT rats showed a decrease in avoidance throughout the tone, compared to eYFP controls (*Figure 3H*, repeated-measures ANOVA, $F_{(1,9)}$=5.23, p=0.034, post-hoc Tukey tests, 6–15 s, all p's < 0.01, 15–18 s, p<0.05). Photosilencing BLA-VS projections did not affect spontaneous bar-pressing (ArchT n = 6, 5.75 average number of presses during laser OFF vs. 7.58 average number of presses during laser ON, $t_{(5)}$=1.27, p=0.259). Overall, these findings suggest that BLA excitation of VS is necessary for the expression of active avoidance.

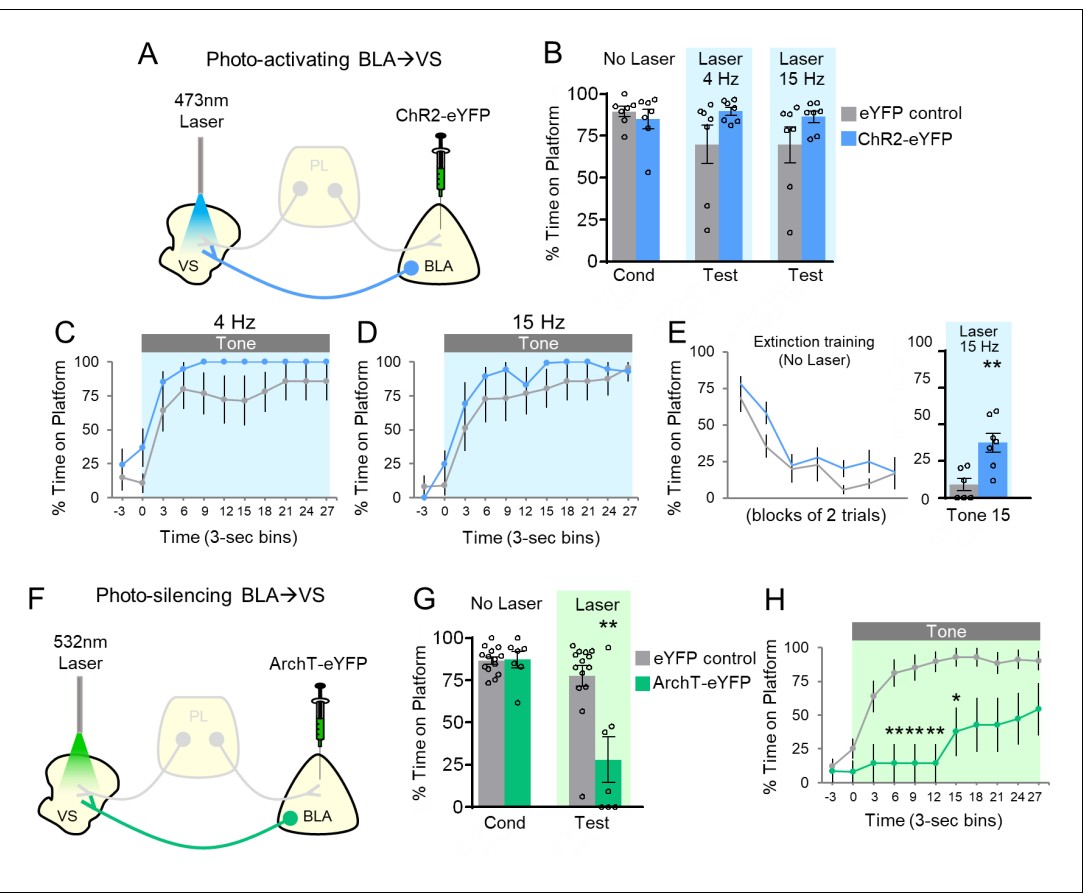

**Figure 3.** Avoidance requires activation of BLA projections to VS. (**A**) Schematic of ChR2 virus infusion and optic probe placement. (**B**) Percent time on platform during the last day of avoidance conditioning (Cond; No Laser), and 4 Hz and 15 Hz Laser tests. (**C**) Timecourse of avoidance during 4 Hz Laser revealed that BLA-VS ChR2 (n = 7) rats showed no significant difference in percent time on platform compared to eYFP (n = 7) controls (repeated-measures ANOVA, post-hoc Tukey). (**D**) Same conventions as in panel C during 15 Hz Laser. (**E**) 15 Hz photoactivation of VS-BLA projections reinstates avoidance following extinction (ChR2 n = 7, eYFP n = 7). (**F**) Schematic of ArchT virus infusion and optic probe placement. (**G**) Percent time on platform during the last day of avoidance conditioning (Cond; No Laser), and Test (Tone with Laser). (**H**) Timecourse of avoidance during Test revealed that ArchT (n = 7) rats showed impaired avoidance compared to eYFP (n = 14) controls (repeated-measures ANOVA, post-hoc Tukey). All data are shown as mean ± SEM. *p<0.05, **p<0.01.

The online version of this article includes the following source data and figure supplement(s) for figure 3:

**Source data 1.** Percent time on platform during manipulation of BLA-VS projections.
**Figure supplement 1.** Spread of AAV expression and location of optic probes.

## Discussion

In this study, we mapped PL outputs that control platform-mediated active avoidance. We found that photoactivation of PL-VS projections impaired avoidance, whereas photoactivation of PL-BLA projections promoted avoidance, suggesting a prefrontal circuit that bidirectionally modulates avoidance behavior. Optogenetic manipulations of BLA-VS projections also revealed that VS receives excitation from the BLA to promote avoidance. These findings add to the growing body of studies showing prefrontal control of active avoidance.

Our results show that photosilencing of the PL-VS circuit had no effect on avoidance, a result which is at odds with our hypothesis that such a manipulation would increase avoidance. One possibility is that a ceiling effect prevented us from observing an increase in avoidance, as we had observed in our initial study when PL somata were photosilenced (see Figure 2E in *Diehl et al., 2018*). A second possibility is that some of the optic probes for our ArchT PL-VS experiments ended

up targeting the shell rather than the core of the VS (see *Figure 1—figure supplement 1B*), possibly suppressing any effect of our PL-VS manipulation. Anatomical studies show that the PL-VS projection targets the core region of the VS, whereas PL-BLA projection targets the shell region of the VS (*Heimer et al., 1997*; *Gorelova and Yang, 1996*; *Zahm, 2000*). Thus, photosilencing PL-VS fibers with the approach we used may have inadvertently photosilenced fibers of passage from PL to BLA, which could cancel out facilitating effects, as photosilencing PL-BLA decreases avoidance. To overcome these challenges, future studies using a recombinase-dependent viral vector to exclusively target PL-VS projections (*Yizhar et al., 2011*) are needed to confirm the effects of photosilencing this projection.

Despite the lack of effect of photosilencing PL-VS projections, we did observe that photoactivating PL-VS projections impaired avoidance. This is in agreement with our previous findings that photoactivation of PL somata impaired avoidance (*Diehl et al., 2018*), and provides indirect evidence that PL inhibition facilitates avoidance. This suggests that the inhibitory tone response in PL targets the VS to drive avoidance. In contrast, we found that excitatory projections from PL to BLA drive avoidance (see model in *Figure 4*). Future studies using PMA could determine if PL neurons showing inhibitory tone responses project to VS, and if PL neurons showing excitatory responses project to BLA by employing photoexcitation-assisted identification of neuronal populations (*Jennings et al., 2013*; *Zhang et al., 2013*; *Ciocchi et al., 2015*; *Beyeler et al., 2016*; *Burgos-Robles et al., 2017*).

Other studies have demonstrated bidirectional control of behavior by frontal areas. PL-VS projections were shown to promote reward-seeking, whereas PL-thalamic projections inhibited reward-seeking (*Otis et al., 2017*). A study of decision making demonstrated that PL dopamine (D2) neurons targeting the BLA signaled the probability of receiving a reward, whereas PL dopamine (D1) neurons targeting the VS signaled the probability of reward omission (*Jenni et al., 2017*). Additionally, a probabilistic reversal learning task investigating the orbitofrontal cortex (OFC) found that OFC-VS projections were necessary for using negative outcomes to guide choices, whereas OFC-amygdala projections were necessary for stabilizing the value of actions (*Groman et al., 2019*). Collectively, these studies demonstrate that bidirectional control of behavior can be governed by specific circuits within PFC, allowing for the selection of the appropriate behavioral response.

A previous study using the shuttle-box avoidance task reported that pre-training lesions in VS enhance avoidance responses (*Lichtenberg et al., 2014*), which disagrees with the current study. However, successful avoidance was measured on the 7[th] day of training after lesions were made, introducing the possibility that other structures compensated for the loss of VS (*Anglada-Figueroa and Quirk, 2005*), thereby forcing an alternative circuit to acquire the task. Moreover, the

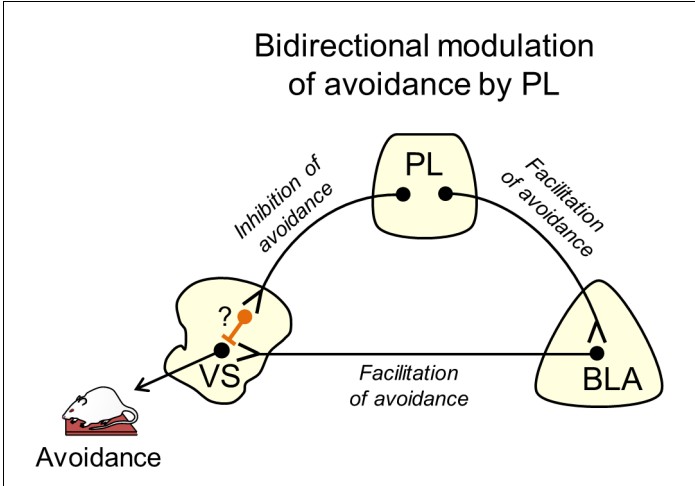

**Figure 4.** Suggested circuit for bidirectional modulation of avoidance by PL. Activity in PL projections to VS decreases avoidance (top left projection). These projections may target fast-spiking interneurons within VS (orange neuron with question mark), which serve to inhibit VS output neurons. Activity in PL projections to BLA increases avoidance (top right projection), as do BLA projections to VS (bottom projection). In this way, PL inputs to VS could gate the impact of BLA inputs on VS output neurons.

lesion approach cannot identify which inputs to VS may be responsible for enhancing avoidance. Another shuttle-box avoidance study showed that disconnecting BLA from VS impaired avoidance (*Ramirez et al., 2015*). Our findings extend this by showing that the activity of BLA terminals within VS is necessary for avoidance. A subset of BLA neurons projecting to VS have been shown to receive inputs from PL (*McGarry and Carter, 2017*), consistent with our model (*Figure 4*). Here, we propose that PL bidirectionally controls active avoidance by modulating the effect of BLA inputs to VS, through activation of BLA and/or feed-forward inhibition of VS, perhaps via fast-spiking interneurons (*Berke, 2011*). In PMA, PL driving of avoidance may be most pronounced during the early part of the tone, as pharmacological inactivation of PL delays but does not prevent avoidance (*Diehl et al., 2019*). This suggests that BLA projections to VS may override feed-forward inhibition from PL to VS, as the tone progresses and there is increased urgency to avoid.

The rodent PL is thought to be homologous to the dorsal anterior cingulate cortex (dACC) in humans (*Bicks et al., 2015*; *Heilbronner et al., 2016*). Neuroimaging studies have found that decreased activity in dACC correlates with the predictability of an aversive stimulus (*Wood et al., 2015*) and the avoidance of a threat (*Aupperle et al., 2015*). In the latter case, decreased activation of dACC was accompanied by increased activation of VS. Another study showed increased coupling between dACC and amygdala during active avoidance (*Collins et al., 2014*). Taken together, these findings demonstrate bidirectional control of active avoidance by dACC and are consistent with PL in rodents guiding appropriate action during the tone (lever pressing vs. moving to the platform).

# Materials and methods

## Key resources table

| Reagent type (species) or resource | Designation | Source or reference | Identifiers | Additional information |
|---|---|---|---|---|
| Genetic reagent (*Chlamydomonas reinhardtii*) | AAV5:CaMKIIα:: hChR2(H134R)-eYFP | UNC vector core | channelrhodopsin (ChR2) | Serotype 5; $4 \times 10^{12}$ particles/mL |
| Genetic reagent (*Aequorea victoria*) | AAV5:CaMKIIα::eYFP | UNC vector core | Enhanced yellow fluorescent protein (eYFP) | Serotype 5; $3 \times 10^{12}$ particles/mL |
| Genetic reagent (*Halorubrum sodomense*) | AAV5:CaMKIIα:: eArchT3.0-eYFP | UNC vector core | Archaerhodopsin (ArchT) | Serotype 5; $4 \times 10^{12}$ particles/mL |
| Software, algorithm | ANY-Maze behavioral software | Stoelting | ANY-Maze | Woodale, IL |

## Subjects

118 adult male Sprague Dawley rats (Envigo Laboratories, Indianapolis, IN) aged 3–5 months and weighing 320–420 g were housed and handled as previously described (*Diehl et al., 2018*). Rats were kept on a restricted diet (18 g/day) of standard laboratory rat chow to facilitate pressing a bar for sucrose pellets (BioServ, Flemington, NJ) on a variable interval schedule of reinforcement (VI-30). Rats were trained until they reached a criterion of >15 presses/min. All procedures were approved by the Institutional Animal Care and Use Committee of the University of Puerto Rico School of Medicine in compliance with the National Institutes of Health guidelines for the care and use of laboratory animals.

## Surgery

Rats were anesthetized with isoflurane inhalant gas (5%) first in an induction chamber, then positioned in a stereotaxic frame (Kopf Instruments, Tujunga, CA). Isoflurane (1–3%) was delivered through a facemask for anesthesia maintenance. Prior to beginning the surgery, rats were administered an analgesic (Meloxicam, 1 mg/Kg) subcutaneously.

For optogenetic experiments, rats were bilaterally infused with viral vectors in the PL PFC(+3.15 mm AP;±0.40 mm ML; −3.60 mm DV to bregma, at a 0°angle) or the BLA ( −2.8 mm AP,±4.8 mm ML, −8.9 mm DV). The syringe was kept inside for an additional 10 min to reduce backflow. Optical fibers (length, 9 mm; 0.22 NA; 200 nm core; constructed with products from Thorlabs, Newton, NJ or purchased preassembled from Newdoon, Hangzhou, Zhejiang, China) were chronically implanted for PL or BLA terminal illumination. For PL terminals, fibers were placed in the VS (+1.2 mm AP,±3.0

mm ML, −3.6 mm DV, at a 15° angle) or the BLA (−2.8 mm AP,±4.8 mm ML, −7.7 mm DV, at a 0° angle). For BLA terminals, fibers were placed in the VS (+1.0 mm AP,±3.0 mm ML, −6.5 mm DV, at a 15°angle). Optical fibers were anchored to the skull with adhesive cement (C and B-Metabond, Parkell, Brentwood, NY; Ortho Acrylic, Bayamón, PR).

After surgery, triple antibiotic was applied topically around the surgery incision, and 24 hr following surgery, an analgesic (Meloxicam, 1 mg/kg) was injected subcutaneously once again. Rats were allowed a minimum of 7 days to recover from surgery prior to behavioral training.

## Behavior

Rats were initially trained to press a bar to receive food pellets on a variable interval reinforcement schedule (VI-30) inside standard operant chambers (Coulbourn Instruments, Whitehall, PA) located in sound-attenuating cubicles (MED Associates, St. Albans, VT). Bar-pressing was used to maintain a constant level of activity against which avoidance could reliably be measured. For platform-mediated avoidance, rats were trained as previously described (Bravo-Rivera et al., 2014). Rats were conditioned with a pure tone (30 s, 4 kHz, 75 dB) co-terminating with a footshock delivered through the floor grids (2 s, 0.3–0.4 mA). The inter-trial interval (ITI) was variable, averaging 3 min. An acrylic square platform (14.0 cm each side, 0.33 cm tall) located in the opposite corner of the sucrose dish allowed rats to be protected from the shock. The platform was fixed to the floor and was present during all stages of training. Rats were conditioned for 10 days with nine tone-shock pairings per day and a VI-30 schedule maintained across all training and test sessions. The availability of food on the side opposite to the platform motivated rats to leave the platform during the ITI, facilitating trial-by-trial assessment of avoidance.

Once platform-mediated avoidance was learned, rats underwent an avoidance expression test, involving the presentation of tones without shock. Laser manipulation occurred during the presentation of tone 1. Following the avoidance expression test, a subset of rats underwent extinction training, which involved 15 tone presentations without shock. Laser manipulation took place on the 15th tone to assess for avoidance reinstatement after extinction.

## Viruses

The adeno-associated viruses (AAVs; serotype 5) were obtained from the University of North Carolina Vector Core (Chapel Hill, NC). Viral titers were $4 \times 10^{12}$ particles/mL for channelrhodopsin (AAV5:CaMKIIα::hChR2(H134R)-eYFP) and archaerhodopsin (AAV5:CaMKIIα::eArchT3.0-eYFP), and $3 \times 10^{12}$ particles/mL control (AAV5:CaMKIIα::eYFP). Rats expressing eYFP in PL or BLA were used to control potential changes in neural activity due to laser-induced overheating of tissue (Stujenske et al., 2015). The CaMKIIα promoter was used to enable transgene expression favoring pyramidal neurons (Liu and Jones, 1996 ) in cortical regions (Jones et al., 1994; Van den Oever et al., 2013; Warthen et al., 2016). Viruses were housed in a −80°C freezer until the day of infusion.

## Laser delivery

Rats expressing channelrhodopsin (ChR2) in PL or BLA were illuminated using a blue diode-pump solid state laser (DPSS, 473 nm, 4 or 15 Hz, 5 ms pulse width, 8–12 mW at the optical fiber tip; Opto-Engine, Midvale, UT), similar to our previous study (Do-Monte et al., 2015). Rats expressing archaerhodopsin (ArchT) in PL were bilaterally illuminated using a DPSS green laser (532 nm, constant, 12–15 mW at the optical fiber tip; OptoEngine). For both ChR2 and ArchT experiments, the laser was activated at tone onset and persisted throughout the 30 s tone presentation. Laser light was passed through a shutter/coupler (200 nm, Oz Optics, Ontario, Canada), patch cord (200 nm core, ThorLabs, Newton NJ), rotary joint (200 nm core, 1 × 2, Doric Lenses, Quebec city, Canada), dual patch cord (0.22 NA, 200 nm core, ThorLabs), and bilateral optical fibers targeting the specific subregions in PL or BLA. Rats were familiarized with the patch cord during bar press training and the last 4 days of avoidance conditioning before the expression test.

## Open field task

Locomotor activity in the open field arena (90 cm diameter) was automatically assessed (ANY-Maze) by measuring distance traveled, time spent, and speed in center or periphery of the arena during 30 s laser off and laser on time periods. A 3 min acclimation period preceded 30 s of 4 Hz photo-

activation, which was followed by an additional 3 min ITI, and a second 30 s laser trial of 15 Hz photo-activation. The distance and speed traveled was used to assess locomotion and time in center was used to assess anxiety.

### Pressing test

Rats pressed on a variable interval reinforcement schedule (VI-30). Changes in the average number of presses were measured during two trials of Laser On (30 s) and Laser Off periods (30 s preceding Laser On). The task began with a 60 s acclimation period, followed by 30 s of Laser On (532 nm), which was followed by a 60 s ITI and 30 s of Laser On. The number of lever activations was compared during Laser Off and Laser On periods within subjects using a paired-t-test.

### Histology

After behavioral experiments, rats were deeply anesthetized with sodium pentobarbital (450 mg/kg i.p.) and transcardially perfused with 0.9% saline followed by a 10% formalin solution. Brains were removed from the skull and stored in 30% sucrose for cryoprotection for at least 72 hr before sectioning and Nissl staining. Histology was analyzed for placement of viral expression and optic fibers.

### Data collection and analysis

Behavior was recorded with digital video cameras (Micro Video Products, Peterborough, Ontario, Canada). ANY-Maze software (Stoelting, Wood Dale, IL) was used to detect the animal's location and movements. ANY-Maze was used to quantify the time spent on the platform during tone presentations as a measure of avoidance. Avoidance to the tone was expressed as a percentage of the 30 s tone presentation. Statistical significance was determined with Student's two-tailed t tests, Fisher Exact tests, Mann–Whitney U tests, or repeated-measures ANOVA, followed by post- hoc Tukey analysis, where appropriate using STATISTICA (Statsoft, Tulsa, OK) and Prism (Graphpad, La Jolla, CA).

## Acknowledgements

This study was supported by NIH grants F32-MH105185 to MMD and R37-MH058883 and P50-MH106435 to GJQ, and the University of Puerto Rico President's Office. This project was also supported by undergraduate training grants R25-GM097635, T34-GM007821, and R25-GM061151. We thank Dr. Anthony Burgos-Robles for comments on the manuscript. We also thank Carlos Rodríguez and Zarkalys Quintero for technical assistance.

## Additional information

### Funding

| Funder | Grant reference number | Author |
| --- | --- | --- |
| National Institute of Mental Health | F32-MH105185 | Maria M Diehl |
| National Institute of Mental Health | R37-MH058883 | Gregory J Quirk |
| National Institute of Mental Health | P50-MH106435 | Gregory J Quirk |
| University of Puerto Rico President's Office | | Gregory J Quirk |
| National Institute of General Medical Sciences | R25-GM097635 | Jorge M Iravedra-Garcia Viviana P Valentín-Valentín |
| National Institute of General Medical Sciences | R25-GM061151 | Fabiola N Gonzalez-Diaz Jonathan Morán-Sierra |
| National Institute of General Medical Sciences | T34-GM007821 | Gabriel Rojas-Bowe |

The funders had no role in study design, data collection and interpretation, or the decision to submit the work for publication.

## Author contributions
Maria M Diehl, Conceptualization, Data curation, Software, Formal analysis, Supervision, Funding acquisition, Validation, Investigation, Visualization, Methodology, Writing - original draft, Project administration, Writing - review and editing; Jorge M Iravedra-Garcia, Gabriel Rojas-Bowe, Data curation, Investigation, Visualization, Writing - review and editing; Jonathan Morán-Sierra, Data curation, Investigation; Fabiola N Gonzalez-Diaz, Investigation; Viviana P Valentín-Valentín, Data curation, Investigation, Writing - review and editing; Gregory J Quirk, Conceptualization, Resources, Formal analysis, Supervision, Funding acquisition, Visualization, Methodology, Writing - original draft, Project administration, Writing - review and editing

## Author ORCIDs
Maria M Diehl https://orcid.org/0000-0002-7370-6106
Jorge M Iravedra-Garcia https://orcid.org/0000-0003-4743-1417
Jonathan Morán-Sierra https://orcid.org/0000-0003-0837-2549
Gabriel Rojas-Bowe http://orcid.org/0000-0002-7042-930X
Viviana P Valentín-Valentín https://orcid.org/0000-0002-5229-2602
Gregory J Quirk http://orcid.org/0000-0002-7534-2764

## Ethics
Animal experimentation: This study was performed in strict accordance with the recommendations in the Guide for the Care and Use of Laboratory Animals of the National Institutes of Health. All of the animals were handled according to approved institutional animal care and use committee (IACUC) protocols (#A3340107) of the University of Puerto Rico. The protocol was approved by the Committee on the Ethics of Animal Experiments of the University of Puerto Rico. All surgery was performed under isofluorane anesthesia, and every effort was made to minimize suffering.

## Decision letter and Author response
Decision letter https://doi.org/10.7554/eLife.59281.sa1
Author response https://doi.org/10.7554/eLife.59281.sa2

# Additional files
## Supplementary files
• Transparent reporting form

## Data availability
All data generated or analysed during this study are included in the manuscript and supporting files.

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
