## [Decision Letter]

**Acceptance summary:**

In this study, the authors significantly extend a prior report showing that inhibitory activity in PL was necessary for PMA behavior, showing that this apparently simple relationship is more complex, involving projections to BLA and VS and also from BLA to VS. Specifically, they show that activating (inhibiting) projections from PL->BLA or from BLA->VS promotes (suppresses) PMA. Further they show that activating projections from PL->VS suppresses avoidance; interestingly inhibiting these projections did not cause increased avoidance.

**Decision letter after peer review:**

Thank you for submitting your article "Divergent projections of the prelimbic cortex bidirectionally regulate active avoidance." for consideration by *eLife*. Your article has been reviewed by two peer reviewers, including Geoffrey Schoenbaum as the Reviewing Editor and Reviewer #1, and the evaluation has been overseen by Michael Frank as the Senior Editor. The following individual involved in review of your submission has agreed to reveal their identity: Michael A McDannald (Reviewer #2).

The reviewers have discussed the reviews with one another and the Reviewing Editor has drafted this decision to help you prepare a revised submission.

Summary:

In this study, the authors follow up a prior report showing that inhibitory activity in PL was necessary for PMA behavior. While the prior report provided evidence using optogenetic manipulations that the novel inhibitory activity was likely involved in the avoidance behavior, the current study extends that to show that this apparently simple relationship is more complex, involving projections to BLA and VS and also from BLA to VS. Specifically, they show that activating (inhibiting) projections from PL->BLA or from BLA->VS promotes (suppresses) PMA. Further they show that activating projections from PL->VS suppresses avoidance; interestingly inhibiting these projections did not cause increased avoidance.

Both reviewers thought the work was elegant in its design and execution and the results were for the most part very clear. The main issue is the discrepancy between the prior result, that inhibition of activity in some PL neurons is necessary to drive PMA, and the failure to clearly demonstrate this effect at terminals downstream in either BLA or VS. This seems to be a key finding that the authors were looking for but did not find, but the discussion and model at the end are presented without commenting on this missing result. This was noted by both reviewers, and on discussion was agreed to be important to address. We are happy to have this addressed with textual changes for now, and the reviews note a variety of reasons this might have occurred. If the authors could consider these and any others they think are worthwhile clearly in the Discussion, and implications for their functional model, that should do it.

Revisions expected in follow-up work:

Some clarification regarding the role of inhibitory activity in PL versus at the terminals in VS, BLA and maybe other areas is needed to resolve the discrepancy described above and in reviews.

Reviewer #1:

In this study, the authors follow up a prior report showing that inhibitory activity in PL was necessary for PMA behavior. While the prior report provided evidence using optogenetic manipulations that the novel inhibitory activity was likely involved in the avoidance behavior, the current study extends that to show that this apparently simple relationship is more complex, involving projections to BLA and VS and also from BLA to VS. Specifically, they show that activating (inhibiting) projections from PL->BLA or from BLA->VS promotes (suppresses) PMA. Further they show that activating projections from PL->VS suppresses avoidance; interestingly inhibiting these projections did not cause increased avoidance.

Generally, these results seem very straightforward and clear. The bidirectional effects are striking, and the work is nicely done and very clean. However I am bothered a bit by what seems to me to be a lack of consistency with the prior report, which is not addressed. Though it is also possible that I simply missed it. Depending on which it is, the authors can likely fix it in the text or straighten me out.

So the issue is this – the prior report claimed that the inhibitory correlates in PL were the unique and prime mover in supporting avoidance. Yet here they fail to find any such effect of inhibition of the downstream terminals. In BLA, inhibition of PL terminals disrupts avoidance and in VS inhibition has no effect. Further they find that activation of output from PL to both areas plays a role in modulating the PMA, which somewhat contradicts the assertation that such activity playing the same role in PMA as in other fear and extinction related procedures.

My own suspicion is that the lack of effects of inhibition in Figure 1F-H is due to a ceiling effect, and that if they had trained the rats more lightly or had extinguished as done in Figures 2 and 3, then they would have found that inhibition of the PL->VS pathway does support PMA. Such data would resolve the seeming inconsistency and, coupled with a more clear discussion of what the effects of activation mean for interpreting the excitatory responses, make more sense to me than the current interpretation, which seems to mostly ignore the issue.

The other possibility to me is that inhibiting across the entire tone is not the same as the neural suppression in some of the neurons at the time of the tone, in which case their approach is not well designed to mimic the correlates.

Of course as I said I may simply not understand how the data all fit together. If this is the case, the authors can clarify it for me. If this is not the case, then I think the failure to find the site where inhibitory PL activity facilitates PMA is a problem for the clean story of bidirectional control the authors want to paint. I am more than happy to have it addressed in the text if the authors want to do this, noting the above caveats and modifying how things are discussed. As I said, I think it is likely an unfortunate ceiling effect. Or the authors could do this experiment if they do not want to fix it textually, but I this is not something I'd require. This is really my only substantive issue with an otherwise very elegant set of results.

Reviewer #2:

This is an elegant series of studies that builds up on Diehl and colleagues prior publication. The rationale for the circuits investigated is clear. The experiments were well designed, executed and interpreted. The results are an important contribution to an emerging literature for the neural circuits supporting avoidance learning.

1) Like the authors, I was a little surprised to see that photoinhibition of PL terminals in the VS did not promote avoidance behavior. I think this null effect warrants more discussion. Looking at the histology in Figure 1—figure supplement 1, it appears that the optical placements for photo-inhibition were ventral to those for photoexcitation. So perhaps photoexcitation more effectively stimulated the accumbens core, whereas photoinhibition targeted the accumbens shell. Also clear is that the PL -> VS projections are extensive. Might it be the case that photoinhibition only targeted a small percentage of PL fibers, a number insufficient to facilitate avoidance? To this second point, the Sabitini lab has developed means of sharpening ferrules to achieve greater light diffusion: https://www.ncbi.nlm.nih.gov/pmc/articles/PMC5533215/.

2) The use of the term ventral striatum is warranted here, as both the PL and BLA target the shell and core subregions of the nucleus accumbens. Moving forward, it will likely be important to acknowledge and examine dissociable roles for the core and shell in avoidance. The authors rightly discuss the Ramirez study examining core. Some discussion of the finding by Lichtenberg et al., 2014 and colleagues is warranted and could be the basis of a more specific discussion of core vs. shell.

[Editors' note: further revisions were suggested prior to acceptance, as described below.]

Thank you for resubmitting your work entitled "Divergent projections of the prelimbic cortex bidirectionally regulate active avoidance." for further consideration by *eLife*. Your revised article has been evaluated by Michael Frank (Senior Editor) and a Reviewing Editor.

The manuscript has been improved but there are some remaining issues that need to be addressed before acceptance, as outlined below:

On discussion, the reviewer's felt that the paper presents enough new data and is enough at odds with the prior report that it would be better presented as a new article than as a straightforward Research Advance follow up. The category of the paper has been changed, but please confirm that this is acceptable in your response.

In addition, it was felt that some additional minor changes to the Discussion were needed to clarify the issue with regard to the specificity of the critical PFC->accumbent manipulation and the role of accumbent in avoidance. The first is that we would like you to state more explicitly that future studies using the recombinant approach would allow targeting of the core projections specifically. This is currently a bit unclear as understanding this requires reference to a phrase several sentences prior stating that the projections are primarily core. Please edit to make this point more easily understood.

The second is at the top of the next paragraph, where we think you could say something more direct regarding the significance of the activation data, assuming you agree. In particular, we feel like the fact that activation impairs avoidance provides indirect evidence c/w the prior findings (that inhibition facilitates). Noting this would help avoid the mis-impression that the data are completely contradictory to the prior study or that accumbent is not involved in avoidance.

Reviewer #1:

The authors have done an excellent job addressing my concerns. I have no further issues.

Reviewer #2:

The authors performed numerous optogenetic experiments, dissecting pathway-specific contributions to avoidance. The most critical experiment, and the reason for submitting as a Research Advance, was inhibition of PL fibers in the nucleus accumbens. Surprisingly, inhibiting PL inputs to the accumbens did not promote avoidance.

The authors reassessment of histology suggests that deferentially targeting the core vs shell may produce diverging effects on avoidance. In combination with the varied ferrule placement, it seems likely that a ceiling effect in avoidance behavior prevented the authors from seeing a facilitation in avoidance behavior.

The advance of the current results depends heavily on the interpretation of the PL to accumbens pathway inhibition. It now seems clear than a rigorous advance will require a new experiment in which the behavioral ceiling is lowered and accumbens core vs. shell are separately and specifically targeted.

---

## [Author Response]

Reviewer #1:In this study, the authors follow up a prior report showing that inhibitory activity in PL was necessary for PMA behavior. While the prior report provided evidence using optogenetic manipulations that the novel inhibitory activity was likely involved in the avoidance behavior, the current study extends that to show that this apparently simple relationship is more complex, involving projections to BLA and VS and also from BLA to VS. Specifically, they show that activating (inhibiting) projections from PL->BLA or from BLA->VS promotes (suppresses) PMA. Further they show that activating projections from PL->VS suppresses avoidance; interestingly inhibiting these projections did not cause increased avoidance.Generally, these results seem very straightforward and clear. The bidirectional effects are striking, and the work is nicely done and very clean. However I am bothered a bit by what seems to me to be a lack of consistency with the prior report, which is not addressed. Though it is also possible that I simply missed it. Depending on which it is, the authors can likely fix it in the text or straighten me out.So the issue is this – the prior report claimed that the inhibitory correlates in PL were the unique and prime mover in supporting avoidance. Yet here they fail to find any such effect of inhibition of the downstream terminals. In BLA, inhibition of PL terminals disrupts avoidance and in VS inhibition has no effect. Further they find that activation of output from PL to both areas plays a role in modulating the PMA, which somewhat contradicts the assertation that such activity playing the same role in PMA as in other fear and extinction related procedures.My own suspicion is that the lack of effects of inhibition in Figure 1F-H is due to a ceiling effect, and that if they had trained the rats more lightly or had extinguished as done in Figures 2 and 3, then they would have found that inhibition of the PL->VS pathway does support PMA. Such data would resolve the seeming inconsistency and, coupled with a more clear discussion of what the effects of activation mean for interpreting the excitatory responses, make more sense to me than the current interpretation, which seems to mostly ignore the issue.The other possibility to me is that inhibiting across the entire tone is not the same as the neural suppression in some of the neurons at the time of the tone, in which case their approach is not well designed to mimic the correlates.

The reviewer is correct that photo-silencing PL projections for 30 sec is not equivalent to mimicking inhibitory tone responses in PL, since there is great variability in the duration of inhibition among these neurons. Rather, we are interested in how the overall inhibition of the PL-VS projection across the entire tone drives avoidance.

Of course as I said I may simply not understand how the data all fit together. If this is the case, the authors can clarify it for me. If this is not the case, then I think the failure to find the site where inhibitory PL activity facilitates PMA is a problem for the clean story of bidirectional control the authors want to paint. I am more than happy to have it addressed in the text if the authors want to do this, noting the above caveats and modifying how things are discussed. As I said, I think it is likely an unfortunate ceiling effect. Or the authors could do this experiment if they do not want to fix it textually, but I this is not something I'd require. This is really my only substantive issue with an otherwise very elegant set of results.

The reviewer is correct to point out that the lack of effect of silencing PL-VS appears to be inconsistent with our original report (Diehl, et al., 2018) and was not dealt with directly in the previous version of the manuscript. We have updated the text in the Discussion to address this point and mention the possibility of: 1) a ceiling effect, 2) fibers of passage from PL to BLA that may have been inadvertently silenced when silencing PL-VS projections, and 3) variation in our optic probe placement affecting our results.

Reviewer #2:This is an elegant series of studies that builds up on Diehl and colleagues prior publication. The rationale for the circuits investigated is clear. The experiments were well designed, executed and interpreted. The results are an important contribution to an emerging literature for the neural circuits supporting avoidance learning.1) Like the authors, I was a little surprised to see that photoinhibition of PL terminals in the VS did not promote avoidance behavior. I think this null effect warrants more discussion. Looking at the histology in Figure 1—figure supplement 1, it appears that the optical placements for photo-inhibition were ventral to those for photoexcitation. So perhaps photoexcitation more effectively stimulated the accumbens core, whereas photoinhibition targeted the accumbens shell. Also clear is that the PL -> VS projections are extensive. Might it be the case that photoinhibition only targeted a small percentage of PL fibers, a number insufficient to facilitate avoidance? To this second point, the Sabitini lab has developed means of sharpening ferrules to achieve greater light diffusion: https://www.ncbi.nlm.nih.gov/pmc/articles/PMC5533215/.

We re-examined the optical probe placements (see Author response image 1 with individual data points labeled, from Figure 1G) and some of these placements indeed targeted the shell, rather than the core for our PL-VS experiments. Rats with optical probes that targeted the shell region showed lower avoidance compared to most of the rats with optical probes targeting the core region. We mention this caveat in the Discussion. We also thank the reviewer for the reference on how to improve our ferrules to achieve greater light illumination for our future studies.

2) The use of the term ventral striatum is warranted here, as both the PL and BLA target the shell and core subregions of the nucleus accumbens. Moving forward, it will likely be important to acknowledge and examine dissociable roles for the core and shell in avoidance. The authors rightly discuss the Ramirez study examining core. Some discussion of the finding by Lichtenberg et al., 2014 is warranted and could be the basis of a more specific discussion of core vs. shell.

The study by Lichtenberg et al., 2014 reported that pre-training lesions in VS enhance avoidance responses in the shuttle task. However, successful avoidance was measured on the 7th day of training after lesions were made, introducing the possibility that other brain structures are likely compensating for the loss of VS (Anglada-Figueroa and Quirk, 2005), thereby forcing an alternative circuit to acquire the task. Moreover, the lesion approach cannot tell us which inputs to VS may be responsible for enhancing avoidance. We now mention this study in the Discussion.

[Editors' note: further revisions were suggested prior to acceptance, as described below.]

On discussion, the reviewer's felt that the paper presents enough new data and is enough at odds with the prior report that it would be better presented as a new article than as a straightforward Research Advance follow up. The category of the paper has been changed, but please confirm that this is acceptable in your response.

We accept to change the article type from Research Advance to a new Research Article.

In addition, it was felt that some additional minor changes to the Discussion were needed to clarify the issue with regard to the specificity of the critical PFC->accumbent manipulation and the role of accumbent in avoidance. The first is that we would like you to state more explicitly that future studies using the recombinant approach would allow targeting of the core projections specifically. This is currently a bit unclear as understanding this requires reference to a phrase several sentences prior stating that the projections are primarily core. Please edit to make this point more easily understood.

We thank the reviewers for these helpful comments in further clarifying the interpretation of our results. We have modified the second paragraph of the Discussion to clarify the point that using a recombinant approach is needed to precisely target PL-VS projections to confirm the effects of photosilencing this projection.

The second is at the top of the next paragraph, where we think you could say something more direct regarding the significance of the activation data, assuming you agree. In particular, we feel like the fact that activation impairs avoidance provides indirect evidence c/w the prior findings (that inhibition facilitates). Noting this would help avoid the mis-impression that the data are completely contradictory to the prior study or that accumbent is not involved in avoidance.

The beginning of the third paragraph is now revised to tie together photoactivation of PL-VS impairing avoidance with our previous finding that photoactivation of PL somata also impaired avoidance.